# Parameter-Efficient Fine-Tuning for Medical Image Analysis: The Missed Opportunity

**Raman Dutt**                                             RAMAN.DUTT@ED.AC.UK
*School of Informatics, The University of Edinburgh, UK*

**Linus Ericsson**                                         LINUS.ERICSSON@ED.AC.UK
*School of Informatics, The University of Edinburgh, UK*

**Pedro Sanchez**                                          PEDRO.SANCHEZ@ED.AC.UK
*School of Engineering, The University of Edinburgh, UK*

**Sotirios A. Tsaftaris**                                  S.TSAFTARIS@ED.AC.UK
*School of Engineering, The University of Edinburgh, UK*

**Timothy Hospedales**                                     T.HOSPEDALES@ED.AC.UK
*School of Informatics, The University of Edinburgh, UK*

**Editors:** Accepted at MIDL 2024

## Abstract

Foundation models have significantly advanced medical image analysis through the *pre-train fine-tune* paradigm. Among various fine-tuning algorithms, Parameter-Efficient Fine-Tuning (PEFT) is increasingly utilized for knowledge transfer across diverse tasks, including vision-language and text-to-image generation. However, its application in medical image analysis is relatively unexplored due to the lack of a structured benchmark for evaluating PEFT methods. This study fills this gap by evaluating 17 distinct PEFT algorithms across convolutional and transformer-based networks on image classification and text-to-image generation tasks using six medical datasets of varying size, modality, and complexity. Through a battery of over 700 controlled experiments, our findings demonstrate PEFT's effectiveness, particularly in low data regimes common in medical imaging, with performance gains of up to 22% in discriminative and generative tasks. These recommendations can assist the community in incorporating PEFT into their workflows and facilitate fair comparisons of future PEFT methods, ensuring alignment with advancements in other areas of machine learning and AI.

**Keywords:** Parameter-Efficient Fine-Tuning, Transfer Learning, Image Classification, Text-to-Image Generation

## 1. Introduction

Medical image analysis has benefited from the deep learning revolution, despite the data-hungry nature of recent foundation models (Dosovitskiy et al., 2021). The challenge of curating large training datasets in medical image analysis is exacerbated due to privacy restrictions, the long-tailed nature of medical conditions of interest, and high annotation cost (Willemink et al., 2020). However, the ability to transfer knowledge from one domain into another (*transfer learning*) has been a key ingredient behind the development of some of the most performant models (Li et al., 2023; Azizi et al., 2021, 2023; Huang et al.; Dutt et al., 2022; Singh and Gorantla, 2020). Under this paradigm, the pre-training is conducted on either out-of-domain non-medical images or unlabeled medical images followed by fine-tuning on in-domain medical images for the specific task. The emergence of 'foundation models' (Bommasani et al., 2021) has further widened the adoption of this approach.

Significant efforts have bolstered the progress in foundation models by scaling them to billions of parameters, hence, the remaining challenge lies in the fine-tuning process that requires striking a delicate balance in adapting the pre-trained model to specialize it for a downstream medical task while avoiding overfitting. This balance has been explored through various fine-tuning algorithms, such as regularized fine-tuning (Xuhong et al., 2018; Gouk et al., 2020). More recently, Parameter-Efficient Fine-Tuning (PEFT) has gained traction (Xie et al., 2023; Rebuffi et al., 2018; Hu et al., 2022; He et al., 2022a). The concept involves freezing the original backbone and fine-tuning either a (very small) existing subset or a small new set of parameters. While the NLP and vision communities have greatly benefitted from structured benchmarks for evaluating PEFT algorithms, a similar direction is lacking in medical image analysis. Furthermore, their efficacy in this domain largely remains underexplored.

In this work, we present the first structured benchmark for evaluating state-of-the-art PEFT algorithms on diverse medical imaging datasets and tasks. Our evaluation compares 16 different techniques across six medical datasets encompassing both CNN and transformer architectures, discriminative diagnosis tasks, and a novel, first-of-a-kind demonstration of PEFT's effectiveness in a generative medical image synthesis task. We experiment with architectures that match the size of recent foundation models introduced for computer vision and medical image analysis (Kirillov et al., 2023; Chambon et al., 2022a). Furthermore, we investigate aspects such as the trade-off between PEFT effectiveness and data volume for the task at hand. We establish the first comprehensive comparison benchmark for PEFT in medical vision and offer the community valuable insights into the, currently, best-suited PEFT methods for different types of tasks.

Our contributions can be summarised by the following questions and their answers:

**Q1:** *How effective is PEFT for low data scenarios?* **A1:** Given a large pre-trained model, benefits from PEFT increase as data volume decreases and model size increases (Sec. 4.1).

**Q2:** *Can PEFT improve transfer to discriminative medical tasks?* **A2:** Yes, three methods achieve consistent gains compared to full fine-tuning, two of which also significantly reduce the computational cost of tuning (Sec. 4.2).

**Q3:** *Can PEFT improve costly text-to-image generation?* **A3:** Yes, PEFT can provide significant performance gains in image generation quality with much lesser computational cost. (Sec. 4.3).

## 2. Related Work

**Finetuning for Medical Image Analysis**. Due to limited availability of data in medical domains, a common paradigm is starting with a deep neural network pre-trained on large natural images, and adapting its weights by fine-tuning (Nima Tajbakhsh, 2016) e.g. via ensembling (Ashnil Kumar, 2017), active learning (Zongwei Zhou, 2017) or with the aid of expert interactions (Guotai Wang, 2018). However, tuning recent large *foundation* models on small datasets – e.g. billions of parameters but only thousands of data points – can cause stability issues and overfitting. Thus, focus has shifted towards what is known as *parameter efficient fine-tuning* (PEFT), i.e. updating only a small number of parameters while keeping the rest fixed.

**PEFT for Medical Image Analysis**. PEFT techniniques can be categorised into three families, adaptive methods (Hu et al., 2022; Rebuffi et al., 2018; Li et al., 2022; Lian et al., 2022), selective methods (Ben Zaken et al., 2022; Frankle et al., 2020; Touvron et al., 2022) and prompt tuning (Lester et al., 2021; Jia et al., 2022; Li and Liang, 2021). A summary of different PEFT methods along with their categorization is given in Table 1. There has been limited adoption of PEFT tech-

| PEFT Method | Paper | Summary | CNNs | ViTs | PEFT Type |
|---|---|---|---|---|---|
| Task-Specific Adapters (TSA) | Li *et al* (Li et al., 2022) | Cross-domain few-shot learning by inserting learnable modules. | ✓ | ✗ | Additive |
| BatchNorm Tuning | Frankle *et al* (Frankle et al., 2020) | Training only BatchNorm layers (even with random initialization) leads to high performance in CNNs. | ✓ | ✗ | Selective |
| Bias Tuning | Cai *et al* (Cai et al., 2020) | Propose TinyTL framework that learns only bias modules. for parameter-efficient on-device learning. | ✓ | ✗ | Selective |
| Scale-Shift Features (SSF) | Lian *et al* (Lian et al., 2022) | Adapt a pre-trained model to downstream datasets by introducing parameters that modulate the extracted features. | ✓ | ✓ | Additive |
| Attention Tuning | Touvron *et al* (Touvron et al., 2022) | Fine-tuning attention layers is sufficient to adapt ViTs to different classification tasks. | ✗ | ✓ | Selective |
| LayerNorm Tuning | Basu *et al* (Basu et al., 2023) | Fine-tuning LayerNorm parameters is a strong baseline for few-shot adaptation. | ✗ | ✓ | Selective |
| BitFit | Zaken *et al* (Ben Zaken et al., 2022) | Fine-tuning the bias terms in a transformer is competitive or better than full-fine-tuning. | ✗ | ✓ | Selective |
| LoRA | Hu *et al* (Hu et al., 2022) | Training injected rank decomposition matrices in transformers is on-par or better than full-fine-tuning. | ✗ | ✓ | Additive |
| AdaptFormer | Chen *et al* (Chen et al., 2022) | Adding lightweight modules increases a ViT's transferability for different image and video tasks. | ✗ | ✓ | Additive |
| SV-Diff | Han *et al* (Han et al., 2023) | Fine-tuning singular values of weight matrices is a parameter-efficient adapter for text-to-image generation models. | U-Net and Text-Encoder in SD | | Additive |
| DiffFit | Xie *et al* (Xie et al., 2023) | Fine-tune only the bias terms and newly-added scaling factors in specific layers. | U-Net and Text-Encoder in SD | | Additive |

Table 1: Summary of the Parameter-Efficient Fine-Tuning (PEFT) methods included in this evaluation, highlighting the specific model type they are designed for and their respective categories.

niques within medical image analysis. In image segmentation, successes have come from learning prompt tokens in a U-Net Ronneberger et al. (2015), or adapters designed specifically for dense prediction tasks (Silva-Rodríguez et al., 2023). On the recently proposed Segment Anything Model (SAM) (Kirillov et al., 2023) — previously unsuccessful in the medical domain — researchers have used PEFT to outperform state-of-the-art methods (Ma et al., 2024; Zhang and Liu, 2023). Finally, PEFT has also been shown to improve fairness in downstream medical tasks (Dutt et al., 2024). **PEFT for Text-to-Image Generation**. Diffusion models (Ho et al., 2020) have led to state-of-the-art results in a variety of tasks such as text-to-image generation (Rombach et al., 2022; Balaji et al., 2022; Saharia et al., 2022), image synthesis (Dhariwal and Nichol, 2021), density estimation (Kingma et al., 2021) and many others. As in other areas, PEFT methods have been proposed to tune these large models. Key approaches include solely tuning bias terms and learnable scaling factors (Xie et al., 2023), attention modules (Moon et al.), adapters (Xiang et al., 2023; Moon et al.) or the singular values of weight matrices (Han et al., 2023).

As of yet, these PEFT methods have not been systematically compared in a medical image analysis setting. We perform the first wide benchmarking study that applies PEFT techniques to diverse tasks in the medical image analysis domain, using state-of-the-art architectures.

## 3. Background

### 3.1. Problem Definition

Let $f$ be a pre-trained model parameterized by $\theta$, $\ell$ be a loss function we wish to minimize and $\mathcal{D} = \{(x_i, y_i)\}_i^N$ be the downstream dataset of interest, consisting of inputs $x_i$ and their targets $y_i$. Starting from the initialization $\theta = \theta_0$, where $\theta_0$ are the weights from pre-training, our objective is then to optimize by gradient descent the total loss $L = \frac{1}{N}\sum_{i=1}^N \ell(f(x_i; \theta), y_i)$. Due to resource constraints, such full fine-tuning is not always possible. It can also be suboptimal to tune the entirety of network weights, as many layers may have learned generally applicable features. Parameter-Efficient Fine-Tuning provides options in these cases, which fall into two broad families. **Selective** methods rely on optimising only a subset of model parameters, $\phi \in \theta$.

This could be a subset of the layers or a specific type of parameter like batch norm. **Additive** methods instead introduce new parameters such that the full set becomes $\theta' = \{\theta, \phi\}$ where $\phi$ can be as simple as a new classifier layer or carefully designed adapters. For both method families, the update rule is $\phi = \phi - \eta \nabla_\phi L$, where $\eta$ is the learning rate.

### 3.2. PEFT Methods For Comparison

We now formally define the different fine-tuning protocols used in the analysis. We begin with a downstream dataset $D$ and a feature extractor $f_\theta$ (pre-trained CNN (ResNet50) or a ViT (Base/Large/Huge)) expected to produce generalizable representations for diverse tasks. First, we freeze all the weights of this feature extractor and enable either an existing subset or a newly added parameter set according to the fine-tuning protocol.

In selective tuning methods, we permit specific parameters to be trainable based on the selected algorithm. For instance, for protocols like BatchNorm and Bias Tuning, the parameters of the 'BatchNorm2d' layers or the 'bias' terms are respectively made trainable. More details including the pseudocode are provided in Appendix (section E).

In **TSA**, our objective is to learn task-specific weights $\phi$ to obtain the task-adapted classifier $f_{(\theta, \phi)}$. Next, we minimize the cross-entropy l oss $L$ over the samples in the downstream dataset $D$ w.r.t the task-specific weights $\phi$. Li et al. (2022) recommend the parallel adapter configuration.

In the **SSF** method, feature modulation is achieved by introducing scale ($\gamma$) and shift ($\beta$) parameters following each operation in the model. The previous operation's output is multiplied by the scale parameter through a dot product and combined with the shift factor. Therefore, for a given input $x$, the output $y$ is calculated using the formula $y = \gamma \cdot x + \beta$.

An **AdaptFormer** module (*AdaptMLP*) consists of two branches wherein the first branch is identical to the MLP block of a vanilla transformer while the second branch consists of a down-projection ($W_{down}$), a ReLU layer, an up-projection ($W_{up}$), and a scaling factor ($s$). The adapted features are combined with the original features entering the *AdaptMLP* block through a residual connection.

**LoRA** is based on the concept that, during adaptation, weight updates exhibit low intrinsic rank. Consequently, when a pre-trained weight matrix $W_0$ is updated, the change ($\Delta W$) is characterized by a low-rank decomposition operation with rank $r$, as shown in Eq. 1 where $B \in \mathbb{R}^{d \times r}$ and $A \in \mathbb{R}^{r \times k}$,

$$W_0 + \Delta W = W_0 + BA. \tag{1}$$

**SV-Diff** performs Singular Value Decomposition (SVD) of the weight matrices of a pre-trained diffusion model and optimizes the spectral shift ($\delta$), defined as the difference between singular values and of the updated and original weight matrix.

## 4. Experiments

### 4.1. How Effective is PEFT For Low Data Scenarios?

**Setup**. We utilized the HAM10000 dataset (Tschandl et al., 2018) and employed three distinct fine-tuning methods, namely Full Fine-tuning, BitFit, and LoRA, in combination with two different encoders, ViT Base and ViT Large. F1-Score was measured at various dataset sizes, commencing with the entire sample size of 7,511 images (100%) and progressively reducing it

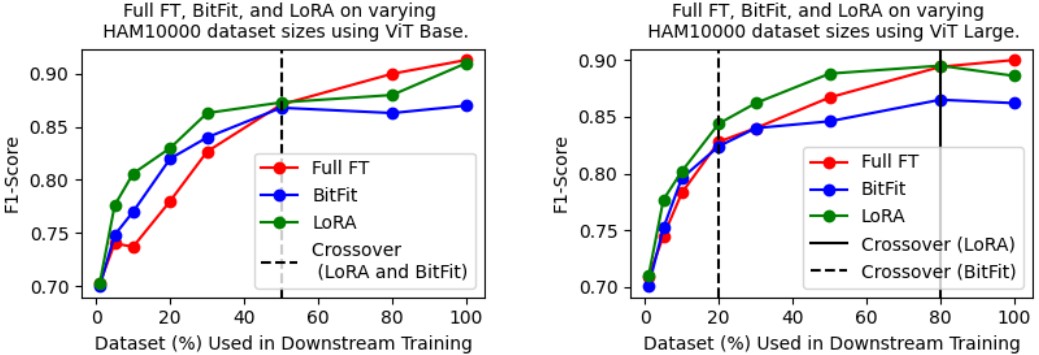

Figure 1: Plots showing the performance comparison for Full Fine-tuning, BitFit and LoRA with varying downstream dataset size for ViT-Base and ViT-Large models.

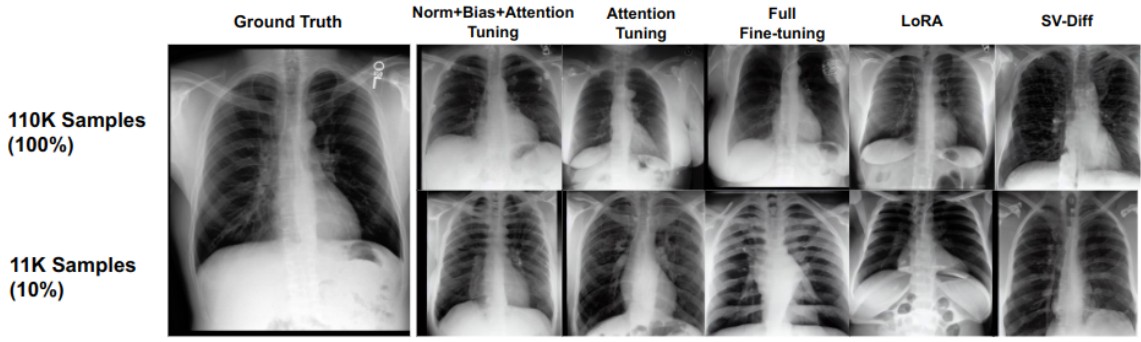

Figure 2: Figure showing text-to-image generation examples with the ground truth in the ascending average rank order (best five) for two data regimes. The input prompt for the generated samples is: "*No acute cardiopulmonary process.*"

to a minimum of 75 images (1%). To account for potential variability in the results, we report the average performance across three random seeds.

**Results**.     The results are shown in Figure 1. For ViT Base (left), we find that when using 100% of available downstream data, full fine-tuning is optimal, closely followed by LoRA. As the availability decreases, however, the benefits from PEFT approaches increase. The crossover is at 50%, when all approaches are approximately equal. For smaller data sizes, both PEFT approaches consistently outperform full FT, with LoRA providing gains of up to 6% over the baseline. For ViT Large, the trend is similar, but the crossover now differs between the PEFT approaches. LoRA overtakes the baseline as early as 80% while BitFit is only better at data volumes below 20%. The take-home message here is that when data are scarce and the upstream model is large, it becomes especially important to consider parameter-efficient tuning.

### 4.2. Can PEFT Improve Transfer to Discriminative Medical Tasks?

**Setup**.     In our discriminative experiments, we use five diverse datasets widely recognized in the medical image analysis community for image classification tasks, BreastUS (Al-Dhabyani et al., 2020), HAM10000 (Tschandl et al., 2018), Fitzpatrick17K (Groh et al., 2021, 2022), Standardized

| Method / Dataset | Full FT (23.5M) | Linear Probing (3.8-7.2K) | TSA (10.6M) | BN Tuning (59.1K) | Bias Tuning (32.7K) | SSF (60.6K) |
|---|---|---|---|---|---|---|
| BreastUS (584) | 0.72±1.1 | 0.61±1.3 | 0.90±0.8 | 0.92±0.9 | 0.89±1.2 | **0.94±0.7** |
| FitzPatrick (5809) | **0.71±0.4** | 0.66±0.8 | 0.69±1.4 | 0.67±1.1 | 0.64±1.3 | **0.71±0.7** |
| HAM10000 (7511) | 0.87±1.2 | 0.82±0.6 | 0.86±1.0 | 0.84±0.6 | 0.70±1.0 | **0.89±0.9** |
| SMDG (9852) | 0.75±0.9 | 0.69±1.0 | **0.85±0.7** | 0.83±1.4 | 0.73±0.6 | 0.84±0.9 |
| Pneumonia (20412) | 0.86±1.4 | 0.80±0.4 | 0.86±1.1 | 0.84±1.5 | 0.85±1.9 | **0.87±1.2** |
| Average F1 Score | 0.77 | 0.72 | 0.83 | 0.82 | 0.76 | **0.85** |
| Average Rank | 2.8 | 5.2 | 2.2 | 3.2 | 4.6 | **1.2** |

Table 2: Comparing different fine-tuning methods for ImageNet pre-trained ResNet50. Dataset size and parameter count are indicated in brackets. The best result for each dataset is highlighted, and the average rank for each fine-tuning method is shown at the bottom.

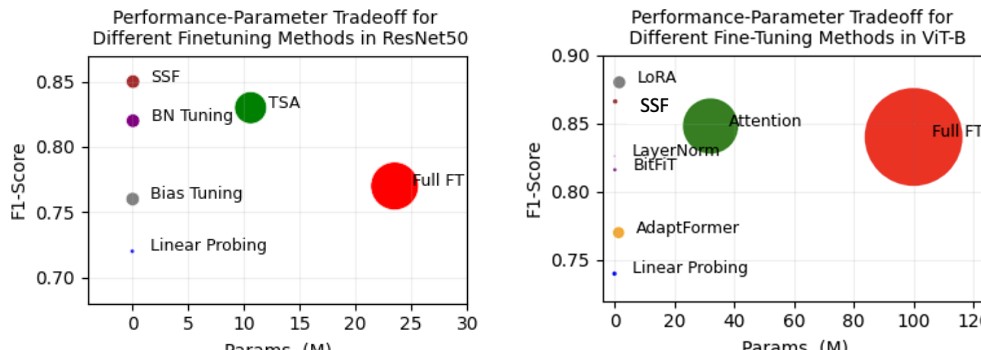

Figure 3: Performance vs. Parameter Count for ResNet50 and ViT-Base Encoders. The marker size indicates the tunable parameter count for each method.

Multi-Channel Dataset for Glaucoma (SMDG) (Kiefer, 2023; Kiefer et al., 2022), and RSNA Pneumonia Detection Dataset (of North America, 2018). The experiments employ ResNet50 (He et al., 2016) and ViT (Base/Large/Huge) (Dosovitskiy et al., 2021) as encoders. All CNN experiments employed ResNet50 pre-trained on ImageNet (Deng et al., 2009) while all ViT variants were pre-trained on ImageNet-21k (Ridnik et al., 2021).

**Results**. We present the results for ResNet-50 in Table 2. Given its convolutional architecture, ResNet-50 is compatible with certain PEFT methods but not others. Overall, full fine-tuning tends to outperform basic linear probing. Observations from the BreastUS and SMDG datasets indicate that most PEFT methods enhance performance beyond the full FT baseline. The SSF method, despite only tuning 60K parameters (0.25%), improves performance by up to 22%. While gains on HAM10000, FitzPatrick and Pneumonia are more modest, the previous section has discussed how these results could potentially vary with changes in data volume and model size. Overall, SSF emerges as the top-performing method based on average F1 score and ranking. Full fine-tuning and TSA present a close tie with the latter emerging on top. BatchNorm and bias tuning perform better than linear probing which turns out to be the worst strategy. Overall, the greatest gains are observed in the smallest dataset (BreastUS), however, the performance

| Encoder | Method / Dataset | Full FT | Linear Probing | Attention Tuning | BitFiT | LoRA | SSF | Adaptformer | LayerNorm Tuning |
|---|---|---|---|---|---|---|---|---|---|
| **ViT Base** | BreastUS (584) | 0.82±1.2 | 0.79±0.7 | 0.93±1.4 | **0.97±1.3** | 0.94±0.6 | 0.95±0.9 | 0.95±0.7 | 0.88±1.1 |
| | FitzPatrick (5,809) | 0.80±1.3 | 0.74±0.6 | 0.76±1.3 | 0.71±1.6 | **0.82±1.4** | 0.77±0.7 | 0.72±1.1 | 0.73±1.2 |
| | HAM10000 (7,511) | **0.91±1.4** | 0.72±0.5 | 0.86±1.2 | 0.87±1.8 | **0.91±1.3** | 0.88±0.8 | 0.76±1.2 | 0.85±1.3 |
| | SMDG (9,852) | 0.80±1.6 | 0.60±0.6 | 0.84±1.8 | 0.66±1.4 | **0.86±1.5** | 0.85±0.9 | 0.60±1.3 | 0.80±1.4 |
| | Pneumonia (20,412) | 0.87±1.7 | 0.86±0.4 | 0.85±1.1 | 0.87±1.2 | 0.86±0.8 | **0.88±1.0** | 0.83±0.9 | 0.87±1.7 |
| | **Average F1 Score** | 0.84 | 0.74 | 0.85 | 0.82 | **0.88** | 0.87 | 0.77 | 0.83 |
| **ViT Large** | BreastUS (584) | 0.84±1.8 | 0.73±0.7 | 0.86±1.3 | **0.95±1.4** | 0.93±1.3 | 0.92±1.8 | **0.95±1.1** | 0.88±1.4 |
| | FitzPatrick (5,809) | **0.82±1.4** | 0.74±0.5 | 0.77±1.2 | 0.74±1.5 | **0.82±1.9** | 0.80±1.3 | 0.72±1.2 | 0.78±1.3 |
| | HAM10000 (7,511) | **0.90±1.6** | 0.82±0.8 | 0.88±1.4 | 0.86±1.1 | 0.89±1.5 | 0.88±1.7 | 0.74±1.0 | 0.87±1.7 |
| | SMDG (9,852) | 0.81±1.5 | 0.77±0.6 | 0.84±1.5 | 0.83±1.9 | 0.83±1.2 | **0.87±1.2** | 0.63±1.3 | 0.85±1.5 |
| | Pneumonia (20,412) | 0.80±1.8 | 0.78±0.9 | 0.81±1.5 | 0.80±1.4 | **0.82±1.1** | 0.80±1.0 | 0.78±1.4 | 0.80±1.6 |
| | **Average F1 Score** | 0.83 | 0.77 | 0.83 | 0.84 | **0.86** | 0.85 | 0.76 | 0.84 |
| **ViT Huge** | BreastUS (584) | 0.92±1.8 | 0.67±0.9 | 0.89±1.5 | **0.96±1.2** | 0.86±1.8 | **0.96±1.1** | 0.93±1.0 | 0.92±1.4 |
| | FitzPatrick (5,809) | 0.69±1.3 | 0.72±0.6 | 0.70±1.3 | 0.72±1.2 | **0.78±1.5** | 0.73±1.1 | 0.72±1.4 | 0.72±0.8 |
| | HAM10000 (7,511) | 0.74±1.7 | 0.74±0.7 | 0.77±1.5 | 0.71±1.4 | **0.87±1.1** | 0.70±0.7 | 0.73±1.0 | 0.72±1.7 |
| | SMDG (9,852) | 0.73±1.5 | 0.64±1.1 | 0.72±1.4 | 0.64±0.9 | **0.83±1.7** | 0.67±1.1 | 0.64±1.2 | 0.67±1.3 |
| | Pneumonia (20,412) | 0.78±1.6 | 0.76±1.3 | 0.78±0.9 | 0.79±1.5 | **0.81±1.7** | 0.79±1.1 | 0.78±1.1 | 0.78±1.2 |
| | **Average F1 Score** | 0.77 | 0.71 | 0.77 | 0.76 | **0.83** | 0.77 | 0.76 | 0.76 |
| **Combined Average Rank** | | 4.1 | 6.7 | 4.5 | 4.5 | **2.4** | 3.1 | 6.0 | 4.7 |

Table 3: Results with different ViT encoders (base/ large/ huge). Dataset size and parameter count are indicated in brackets. The best result for each dataset is highlighted, and the average rank for each fine-tuning method is shown at the end. Parameter count for each PEFT method and encoder is presented in Appendix Sec. D.

gap between full fine-tuning and PEFT methods minimizes with an increase in dataset size. For **Transformer** models in Tab. 3, the situation is similar. The biggest gains over full FT are on BreastUS and SMDG, while linear probing underperforms here as well. The best PEFT method is **LoRA**, for both average F1 score and rank, across all five datasets. AdaptFormer does not perform well and even falls behind linear probing for ViT Large. This can be attributed to the fact that this method was mainly designed for video recognition tasks. We also see that the benefits of PEFT increase slightly as the model size increases, with a 4% improvement for ViT Base going to 6% for ViT Huge. This is an interesting finding, and agrees with Sec. 4.1, as the proportion of parameters tuned actually decreases for the larger models.

Figure 3 illustrates the trade-off between each method's performance and parameter count. This comparison is crucial as different applications may prioritize either superior performance or computational efficiency. For the results produced by the ResNet50 (shown on the left), each PEFT method lies on the Pareto frontier, indicating that a specific method could be selected based on the prioritization of either performance or cost. Remarkably, the SSF method stands out by delivering high performance at a significantly reduced cost. In the case of the ViT-B model, LoRA emerges as the prominent choice, outpacing SSF while maintaining a similar computational expense.

To answer our question *can PEFT Improve Transfer to Discriminative Medical Tasks?* Yes, **TSA, SSF and LoRA** provide consistent improvements over full fine-tuning while requiring as little as 0.25% of parameters.

### 4.3. Can PEFT Improve Costly Text-to-Image Generation?

**Setup**. We use the MIMIC-CXR dataset (v. 2.0.0) (Johnson et al., 2019). Following the recommendations of Chambon et al. (2022b), we fine-tune only the U-Net component (keeping text-encoder and VAE frozen) of the stable diffusion pipeline for different sizes of the downstream dataset (110K, 55K, and 11K, representing 100%, 50% and 10% of the entire dataset). For analysis, we compare the full-finetuning of U-Net with 7 different PEFT methods and report the

| FID (↓) \ PEFT | Full FT (85.9M) | Attention (26.7M) | Bias (0.34M) | Norm (0.2M) | Bias+Norm+Attention (26.7M) | LoRA (0.8M) | SV-Diff (0.22M) | DiffFit (0.58M) |
|---|---|---|---|---|---|---|---|---|
| FID @ 110K | 58.74 | 52.41 | **20.81** | 29.84 | 35.93 | 439.65 | 23.59 | 42.50 |
| FID @ 55K | 98.48 | 39.76 | 28.67 | 29.24 | 62.34 | 392.45 | **22.06** | 51.24 |
| FID @ 11K | 74.70 | 61.01 | 17.87 | 37.30 | 43.46 | 399.28 | 27.02 | **17.49** |
| Average FID (↓) | 77.30 | 51.06 | **22.45** | 32.12 | 47.24 | 410.46 | 24.22 | 37.07 |
| Average Rank | 7 | 5.33 | **1.67** | 3.33 | 5 | 8 | 2 | 3.67 |

Table 4: Table presenting the FID scores for different strategies of fine-tuning the U-Net sub-component on different ratios of the MIMIC dataset. Full Fine-tuning is outperformed by almost every other method by a significant margin.

FID Score over 1000 test images averaged across four random seeds. Stable Diffusion pipelines and PEFT methods were implemented using the *diffusers* (von Platen et al., 2022) and *peft* (Sourab Mangrulkar, 2022) packages.

**Results**. Refer to Table 4 for quantitative results and Figure 2 for example images generated using different fine-tuning methods for two scenarios (110K and 11K samples). **Note** that certain PEFT strategies (bias tuning, norm tuning, etc) have not been published in the literature in the context of text-to-image generation but are included here in experiments.

For all data volumes, several PEFT methods outperformed full fine-tuning with significant differences in FID scores. A particularly interesting observation is that simple strategies such as fine-tuning just the bias or normalization layers are amongst the best performers, assuming first and third ranks respectively. Other PEFT methods designed exclusively for text-to-image generation tasks (SV-Diff and DiffFit) follow closely and also outperform full fine-tuning. Interestingly, LoRA, the best-performing method for classification tasks fails to provide any benefits in image generation. Overall, PEFT shows strong promise in improving the medical image generation quality across different data volumes.

## 5. Conclusion

We performed the first, thorough evaluation of parameter-efficient fine-tuning for the medical image analysis domain covering a wide range of algorithms, architectures, datasets, and tasks. **For discriminative tasks,** the benefits of PEFT increase with decreasing data volume and increasing model size. Furthermore, The benefits of PEFT are especially prominent for low to medium-scale datasets, which are particularly common in the medical domain. SSF and LoRA emerged as the best-performing methods for CNNs and ViTs respectively in our analysis. **For generative tasks,** simple strategies such as Bias Tuning and tailored methods such as SV-Diff provide significant performance gains over conventional strategies. With rapid progress in studying efficient fine-tuning algorithms, this benchmark would allow easy integration and evaluation of new PEFT methods on diverse medical tasks in future.

## Acknowledgments

Raman Dutt is supported by the United Kingdom Research and Innovation (grant EP/S02431X/1), UKRI Centre for Doctoral Training in Biomedical AI at the University of Edinburgh, School of Informatics.

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

## Appendix A. Results on Self-Supervised Encoders

We extended our evaluation to include ViT encoders (ViT Base) pre-trained using different self-supervised objectives. More specifically, we adopted the highly-effective Masked Autoencoder (MAE) (He et al., 2022b) and Contrastive Language-Image Pretraining (CLIP) (Radford et al., 2021) strategies.

Our results align with our previous observations outlined in section 4.2. **LoRA** continues to outperform other PEFT methods across both pre-training objectives. In the case of MAE ViT, Attention Tuning performs slightly better than SSF. Overall, the average ranks are very similar to the ones originally reported in the paper.

| Encoder | Dataset | Full FT | Linear Readout | Attention Tuning | BitFiT | LoRA | SSF | Adaptformer | LayerNorm Tuning |
|---|---|---|---|---|---|---|---|---|---|
| | BreastUS | 0.89 | 0.80 | 0.94 | 0.84 | **0.97** | 0.95 | 0.84 | 0.92 |
| | FitzPatrick | 0.77 | 0.68 | **0.80** | 0.72 | 0.78 | 0.79 | 0.72 | 0.73 |
| ViT Base MAE | HAM10000 | 0.83 | 0.68 | **0.90** | 0.80 | 0.87 | 0.81 | 0.70 | 0.85 |
| | SMDG | **0.87** | 0.72 | 0.86 | 0.78 | **0.87** | 0.85 | 0.75 | 0.82 |
| | Pneumonia | **0.87** | 0.83 | 0.86 | **0.87** | **0.87** | 0.83 | 0.86 | 0.85 |
| | **Average Rank** | 3.0 | 7.8 | 2.4 | 5.2 | **2.2** | 4.2 | 6.6 | 4.6 |
| | BreastUS | 0.91 | 0.83 | 0.94 | 0.91 | 0.94 | **0.97** | 0.91 | 0.95 |
| | FitzPatrick | **0.82** | 0.69 | 0.80 | 0.72 | 0.81 | 0.78 | 0.72 | 0.78 |
| ViT Base CLIP | HAM10000 | 0.84 | 0.77 | 0.85 | 0.81 | **0.89** | 0.87 | 0.81 | 0.87 |
| | SMDG | 0.83 | 0.69 | 0.84 | 0.82 | **0.88** | 0.87 | 0.76 | 0.85 |
| | Pneumonia | 0.86 | 0.8 | 0.86 | 0.85 | **0.87** | **0.87** | 0.84 | 0.85 |
| | **Average Rank** | 3.6 | 8.0 | 3.4 | 5.8 | **1.8** | 2.8 | 7.0 | 3.6 |

Table 5: Table presenting the results for ViT Base model pre-trained using different self-supervised objectives.

## Appendix B. Training Details

**Details on Batch Size and Optimizer:** For each experiment, we used a batch size of **512** and **AdamW** optimizer (Loshchilov and Hutter, 2017). Our initial experiments concluded that the choice of optimizer does not have any major impact on the downstream performance and hence, we proceeded with AdamW as it is one of the most commonly adopted optimizers for both discriminative and generative tasks.

**Details on Learning Rate Selection:** We observed that fine-tuning of PEFT methods shows a preference for larger learning rates (about a magnitude higher than the full fine-tuning). However, since each fine-tuning strategy, model architecture, and dataset might benefit from a different learning rate, we relied on a common HPO procedure, implemented using the *Optuna* package (Akiba et al., 2019), to obtain the optimal learning rate for each competitor, in order to perform a fair comparison. The goal of the HPO was to find the best learning rate by maximizing the performance on the validation set. We ran the HPO procedure to find the optimal learning rate for each fine-tuning strategy, model architecture and dataset. Finally, we used the HPO-recommended learning rates and reported the performance on the test set.

## Appendix C. Update Rules for PEFT Algorithms

### C.1. Task-Specific Adapters (TSA)

In **TSA**, our objective is to learn task-specific weights $\phi$ to obtain the task-adapted classifier $f_{(\theta,\phi)}$. Next, we minimize the cross-entropy loss $L$ over the samples in the downstream dataset $D$ w.r.t the task-specific weights $\phi$. Li et al. (2022) recommend the parallel adapter configuration. The output of the $l$-th layer of the feature extractor $f_\theta$ can be combined with the task-specific adapters $r_\phi$ for an input tensor $h \in \mathbb{R}^{W \times H \times C}$ in a parallel configuration using,

$$f_{(\theta_l,\phi)}(h) = r_\phi(h) + f_{\theta_l}(h). \tag{2}$$

### C.2. Adaptformer

In *Adaptformer* (section 3.2), the adapted features are obtained using equation 3. These features are then combined with the original features entering the *AdaptMLP* block through a residual connection, described in equation 4. Here, *ReLU* and *LN* describe the Rectified Linear Unit and Layer Normalization respectively.

$$x_{adap} = ReLU(LN(x_{orig}) \cdot W_{down}) \cdot W_{up} \tag{3}$$

$$x_{final} = MLP(LN(x_{orig})) + s.x_{adap} + x_{orig} \tag{4}$$

### C.3. SV-Diff

**SV-Diff** performs Singular Value Decomposition (SVD) of the weight matrices of a pre-trained diffusion model (Eq. 5) and optimizes the spectral shift ($\delta$), defined as the difference between singular values and of the updated and original weight matrix.

The update rule is defined in Eq. 6,

$$W = U\Sigma V^\intercal \quad \text{with} \quad \Sigma = \text{diag}(\sigma), \tag{5}$$

$$W_\delta = U\Sigma_\delta V^\intercal \quad \text{with} \quad \Sigma_\delta = \text{diag}(\text{ReLU}(\sigma + \delta)). \tag{6}$$

### C.4. DiffFit

DiffFit builds on the BitFit approach (Ben Zaken et al., 2022) and fine-tunes only the bias, normalization terms and the class-condition module. Further, learnable scaling factors $\gamma$ are introduced. A minimal implementation protocol of DiffFit is provided in Section E.2.4.

## Appendix D. Trainable Parameter Count for PEFT Methods

The trainable parameter count for each PEFT method and ViT variant is presented in Table 6. For *Linear probing*, the parameter count depends on the number of classes in the downstream dataset. Certain methods such as *Attention Tuning*, despite of falling under the PEFT, show a high parameter count. For other PEFT methods, the number of trainable parameters do not grow as rapidly as the total parameter in the respective ViT variant.

| Encoder | Full FT | Linear Probing | Attention Tuning | BitFit | LoRA | SSF | Adaptformer | LayerNorm Tuning |
|---------|---------|----------------|------------------|--------|------|-----|-------------|------------------|
| ViT Base | 87.2 M | 3.8 - 7.2 K | 28.5 M | 0.1 M | 0.6 M | 0.2 M | 0.1 M | 0.04 M |
| ViT Large | 303 M | 3.8 - 7.2 K | 100 M | 0.2 M | 1.5 M | 0.5 M | 0.3 M | 0.1 M |
| ViT Huge | 630 M | 3.8 - 7.2 K | 210 M | 0.4 M | 2. 6 M | 0.9 M | 0.5 M | 0.2 M |

Table 6: Table presenting the trainable parameter count for each PEFT method and ViT variant (Base/ Large/ Huge)

## Appendix E. Training Protocols of Selective PEFT Methods

### E.1. Discriminative Tasks

#### E.1.1. Normalization Tuning (CNNs)

```python
def set_module_grad_status(module, flag=False):
    if isinstance(module, list):
        # print("list", module)
        for m in module:
            set_module_grad_status(m, flag)
    else:
        # print("not a list", module)
        for p in module.parameters():
            p.requires_grad = flag

# Function to enable batchnorm parameters
def enable_bn_update(model):
    for m in model.modules():
        if type(m) in [nn.BatchNorm2d, nn.GroupNorm]:
            if m.weight is not None:
                set_module_grad_status(m, True)
```

Code Listing 1: Fine-Tuning only the normalization parameters (BatchNorm) in CNNs

#### E.1.2. Bias Tuning (CNNs)

```python
def enable_bias_update(model):
    for m in model.modules():
        for name, param in m.named_parameters():
            if name == "bias":
                param.requires_grad = True
```

Code Listing 2: Fine-Tuning only the bias parameters in CNNs

#### E.1.3. Attention Tuning (ViTs)

```python
def tune_attention_layers(model, model_type):

    for name_p,p in model.named_parameters():
        if '.attn.' in name_p or 'attention' in name_p:
            p.requires_grad = True
        else:
            p.requires_grad = False

        model.head.weight.requires_grad = True
        model.head.bias.requires_grad = True

        # POSITION EMBEDDING
        try:
            model.pos_embed.requires_grad = True
        except:
            print('no pos embedding')
```

```
17
18          # PATCH EMBEDDING
19          try:
20              for p in model.patch_embed.parameters():
21                  p.requires_grad = False
22          except:
23              print('no patch embed')
```

Code Listing 3: Fine-Tuning only the attention parameters in ViTs

### E.1.4. TASK-SPECIFIC ADAPTERS (TSA)

```
1  # orig_resnet = pretrained ResNet
2
3  for block in orig_resnet.layer1:
4      for name, m in block.named_children():
5          if isinstance(m, nn.Conv2d):
6              new_conv = conv_tsa(m, self.ad_type)
7              setattr(block, name, new_conv)
8
9  for block in orig_resnet.layer2:
10     for name, m in block.named_children():
11         if isinstance(m, nn.Conv2d):
12             new_conv = conv_tsa(m, self.ad_type)
13             setattr(block, name, new_conv)
14
15 for block in orig_resnet.layer3:
16     for name, m in block.named_children():
17         if isinstance(m, nn.Conv2d):
18             new_conv = conv_tsa(m, self.ad_type)
19             setattr(block, name, new_conv)
20
21 for block in orig_resnet.layer4:
22     for name, m in block.named_children():
23         if isinstance(m, nn.Conv2d):
24             new_conv = conv_tsa(m, self.ad_type)
25             setattr(block, name, new_conv)
```

Code Listing 4: Attaching TSA layers to a pre-trained ResNet

### E.2. Generative Tasks

### E.2.1. NORM TUNING

```
1  def enable_norm_update(model):
2      print("Enabling Normalization layers")
3      for m in model.modules():
4          for name, param in m.named_parameters():
5              if "norm" in name:
6                  param.requires_grad = True
```

Code Listing 5: Fine-Tuning only the normalization parameters in Stable Diffusion (U-Net)

### E.2.2. Bias Tuning

```
1 def enable_bias_update ( model ):
2     print ("Enabling Bias layers")
3     for m in model.modules ():
4         for name , param in m.named_parameters ():
5             if name == "bias":
6                 param.requires_grad = True
```

Code Listing 6: Fine-Tuning only the bias parameters in Stable Diffusion (U-Net)

### E.2.3. Bias Tuning

```
1 def enable_attention_update ( model ):
2     print ("Enabling Attention layers")
3     for m in model.modules ():
4         for name , param in m.named_parameters ():
5             if "attentions" in name:
6                 param.requires_grad = True
```

Code Listing 7: Fine-Tuning only the attention parameters in Stable Diffusion (U-Net)

### E.2.4. DIFFFIT

```
1 def enable_difffit_update ( model: nn.Module ):
2
3     trainable_names = ["bias","norm","gamma","y_embed"]
4
5     for par_name , par_tensor in model.named_parameters ():
6         par_tensor
    .requires_grad = any ([ kw in par_name for kw in trainable_names ])
7
8     return model
```

Code Listing 8: Fine-Tuning protocol for DiffFit

