# OpenReview forum: "Parameter-Efficient Fine-Tuning for Medical Image Analysis: The Missed Opportunity"
_MIDL.io/2024/Conference — MIDL 2024 Oral_

### Official Review · Reviewer_ZQbg · 2024-02-20

**Confidence:** 4
**Preliminary Rating:** 4
**Final Rating:** 4

**Summary:**

This paper proposes the first extensive benchmark of Parameter-Efficient Fine-Tuning (PEFT) methods on diverse medical imaging datasets and tasks. Through these experiments, the paper attempts to answer several keys practical keys questions such as:
- How well PEFT methods are effective when considering low data regimes and large models?
- Are PEFT methods able to improve transfer to medical discriminative tasks?
- Can PEFT Improve Costly Text-to-Image Generation?

**Strengths:**

- The paper is well-written and is easy to follow.
- The paper provides an exhaustive list of existing PEFT methods while evaluating them under different datasets, tasks, architectures, and data regimes.
- Since medical imaging faces different challenges than natural image datasets, I think this benchmark and the obtained findings covering many settings can serve as strong guidelines for the community to adapt large pretrained models for medical tasks.

**Weaknesses:**

I don't have major concerns. Even though I know that experiments cannot be carried out for all datasets and tasks, dense tasks such as object detection, tracking, semantic segmentation, and instance segmentation are extremely popular in the medical imaging community. Therefore it would have been very interesting to see how well large pretrained models can be adapted with PEFT methods for some of these dense tasks. I think it would make the paper even more helpful for the community.

**Detailed Comments:**

No specific suggestions or clarifications required

**Justification Of Final Rating:**

Given the limited rebuttal Time, I understand that authors could not run the experiment even. I will keep my grade as it is (weak accept) but I think this work is worth being published and would be useful for the community.

**Justification Of The Preliminary Rating:**

Since medical imaging faces different challenges than natural image datasets, I think this benchmark and the obtained findings covering many settings can serve as strong guidelines for the community to adapt large pretrained models for medical tasks.

**Questions To Address In The Rebuttal:**

Please address the concerns from the weaknesses section.

---

> ### Author Response · Authors · 2024-03-12
> **Addressing only Classification and Generative Tasks**
>
> Thank you for your time and effort in providing an insightful review and recognizing our work as _well-written_ and _exhaustive_. We have addressed your comments below.
>
> **Addressing only classification and generative tasks:** We agree it would be interesting to include detection and segmentation in our analysis and expect that PEFT methods will also benefit dense medical imaging tasks as they have benefitted dense tasks outside medical imaging [1]. However, our broader goal was to evaluate the two main categories of tasks: _discriminative_, through image classification and _generative_, through text-to-image generation. Nevertheless, given the constraints on space and author feedback time, we must leave this for a future extension of this work.
>
> **References**
> 1. Chen, Hao, et al. "Conv-adapter: Exploring parameter efficient transfer learning for convnets." arXiv preprint arXiv:2208.07463 (2022).

---

> > ### Comment · Reviewer_ZQbg · 2024-03-21
> >
> > Given the limited rebuttal Time, I understand that authors could not run the experiment I wanted. I will keep my grade as it is (weak accept)

---

### Official Review · Reviewer_7zWN · 2024-02-20

**Confidence:** 4
**Preliminary Rating:** 5
**Recommendation:** Oral
**Final Rating:** 5

**Summary:**

This paper explores the use of Parameter-Efficient Fine-Tuning (PEFT) methods for medical image analysis, and in particular, for image classification and text-to-image generation tasks. This work evaluates several popular PEFT methods for different tasks, under several data regimes, adapting models pre-trained on large natural image datasets.

**Strengths:**

- The paper is well-structured and easy to follow.
- The motivation of the paper is sound.
- The open questions and conclusions that drive the paper are well-known in natural image analysis. However, they have hardly been investigated in the context of medical image analysis, so this paper could draw attention to the PEFT setting and serve as a good starting point for future benchmarks.
- The selected PEFT methods and backbones are appropriate. The authors present a wide umbrella of appealing experiments, which demonstrate the potential of PEFT in medical image analysis.

**Weaknesses:**

- Although the authors talk about SAM, and some works on segmentation, only image classification and image generation tasks are addressed.
- The core idea of PEFT implies re-using rich pre-trained features and adapting them to downstream tasks in an efficient fashion. Under this scope, using features pre-trained on natural images supposes a large domain shift to the target domain. For some of the target tasks, there already exist domain-specific pre-trained models (e.g. MedCLIP [a] for Chest X-Ray), which may provide a more data- and parameter-efficient adaptation.

[a] Wang, Zifeng and Wu, Zhenbang and Agarwal, Dinesh and Sun, Jimeng. (2022). MedCLIP: Contrastive Learning from Unpaired Medical Images and Texts. EMNLP'22.

**Detailed Comments:**

No additional comments.

**Justification Of Final Rating:**

I understand that the length limitations of the article do not allow providing a more detailed analysis of PEFT methods in other medical imaging applications, such as image segmentation, or the use of domain-specific pre-trained networks. I think the paper does provide enough experimental evidence to show the potential of PEFT on medical image analysis and constitutes a considerable contribution to the current literature. Therefore, I will keep the preliminary rating.

**Justification Of The Preliminary Rating:**

The paper assesses a setting largely unexplored, and potentially impactful, to the medical image analysis community. The authors show experiments with convincing backbone architectures, PEFT adaptation strategies, target tasks, and results.

**Questions To Address In The Rebuttal:**

Please, see Weaknesses.

**Special Issue:**

Yes

---

> ### Author Response · Authors · 2024-03-12
> **Evaluating Domain-specific Pre-trained Models and Benchmarking only Classification and Generative Tasks**
>
> Thank you for your time and feedback for providing this review. We have addressed the comments below.
>
> **Evaluating Domain-specific Pre-trained Models:** We agree that exploring domain-specific pre-trained models would be interesting, and could potentially improve the results further. However, the majority of studies today have used ImageNet1K or ImageNet21K as pre-training data, so we used these pre-trained models to be in line with the most common practice of prior work. We also already presented additional results on using different (and stronger than supervised learning) pre-training schemes including MAE [1] and CLIP [2] in the Appendix. Our combined results (Tables 2,3 and 5) show the dominance of PEFT strategies across different pre-training schemes. Hence, we believe that we would see a similar pattern for domain-specific pre-training objectives as well. However, given the limited author feedback window and paper space, we must leave this to future work.
>
> **Addressing only classification and generative tasks:** We agree this would be interesting to know and expect that PEFT methods will also benefit dense medical imaging tasks as they have benefitted dense tasks outside of medical imaging [3]. Our broader goal was to evaluate the two main categories of tasks: _discriminative_, through image classification and _generative_, through text-to-image generation. Nevertheless, given the constraints on space and author feedback time, we must leave this for a future extension of this work.
>
> **References**
> 1. He, Kaiming, et al. "Masked autoencoders are scalable vision learners." Proceedings of the IEEE/CVF conference on computer vision and pattern recognition. 2022.
>
> 2. Radford, Alec, et al. "Learning transferable visual models from natural language supervision." International conference on machine learning. PMLR, 2021.
>
> 3. Chen, Hao, et al. "Conv-adapter: Exploring parameter efficient transfer learning for convnets." arXiv preprint arXiv:2208.07463 (2022).

---

> > ### Comment · Reviewer_7zWN · 2024-03-22
> >
> > Thanks for the detailed response!
> >
> > I do not fully agree with the claim that pre-training schemes such as MAE and CLIP are necessarily better than supervised pre-training. Supervised pre-training is yet an ongoing evolving field, and if properly done, it is largely competitive, according to recent literature [1,2]. Nevertheless, although being an interesting discussion theme, this topic does not directly affect the paper under review.
> >
> > [1] Bulent et al. "No Reason for No Supervision: Improved Generalization in Supervised Models". ICLR 2023.
> >
> > [2] Wang et al. "Revisiting the Transferability of Supervised Pretraining: an MLP Perspective". CVPR 2022.
> >
> > I understand that the length limitations of the article do not allow providing a more detailed analysis of PEFT methods in other medical imaging applications, such as image segmentation, or the use of domain-specific pre-trained networks. I think the paper does provide enough experimental evidence to show the potential of PEFT on medical image analysis and constitutes a considerable contribution to the current literature. Therefore, I will keep the preliminary rating.

---

### Official Review · Reviewer_mQqj · 2024-03-04

**Confidence:** 4
**Preliminary Rating:** 4
**Final Rating:** 5

**Summary:**

This paper aims to build a benchmark comparison of different fine-tuning (FT) strategies for medical image classification and synthesis, using both convolutional and transformer backbones. Comprehensive fine-tuning experiments were conducted to identify the best-performing Parameter-Efficient Fine-Tuning (PEFT) strategies for different tasks.

**Strengths:**

- Comprehensive comparison experiments were deployed using various datasets.
- Clear conclusions were drawn based on the results, helping readers determine which PEFT methods performed better on specific tasks.
- Good visualizations such as Fig. 3 were displayed.

**Weaknesses:**

- Some training details were missing, such as the batch size and optimizer used. Were the same batch size and optimizer used for different PEFTs?
- In each dataset, only FT-based methods were trained and compared without considering SOTA methods for each dataset, such as those without pre-training but with an appropriate model and parameter size.

**Detailed Comments:**

See comments above

**Justification Of Final Rating:**

The authors will add more details regarding the experimental hyper parameters. The authors also emphasize the focus of this work as evaluating various FT methods rather than outperforming SOTA methods. I decide to increase my rating to '5: strong accept'.

**Justification Of The Preliminary Rating:**

Even though training details for each PEFT method and SOTA comparisons in each dataset were missing, this work does have merit in providing a benchmark comparison of PEFT methods for medical image analysis tasks and contributing to the field.

**Questions To Address In The Rebuttal:**

Please add training details for the different methods used in the study. Also, discuss SOTA methods in each dataset and how the best performing PEFTs compare to these methods.

---

> ### Author Response · Authors · 2024-03-12
> **Training Details and Comparison with SoTA**
>
> Thank you for the insightful review and for identifying our paper as a _comprehensive analysis_ supported with _good visualizations_ and _clear conclusions_. We have addressed your comments below.
>
> **Training Details:** For each experiment, we used the same batch size of **512** and optimizer **AdamW** [1]. From our initial experiments, we observed that the choice of optimizer does not affect the fine-tuning performance and hence, we selected one of the most widely adopted optimizers for both classification and generative tasks.
>
> **Additional Details:** We observed that fine-tuning of PEFT methods shows a preference for larger learning rates (about a magnitude higher than the full fine-tuning). However, since each fine-tuning strategy, model architecture, and dataset might benefit from a different learning rate, we relied on a common HPO procedure to obtain the optimal learning rate for each competitor, in order to perform a fair comparison. We will add further details about the HPO procedure in the Appendix of the final version of the paper.
>
> **Comparison with SoTA methods:** The primary objective of this study was to evaluate and compare various fine-tuning strategies, revealing the effectiveness of PEFT methods specifically tailored for medical tasks. While prior studies may have employed diverse techniques such as advanced architectures, regularizers or augmentations to achieve state-of-the-art (SoTA) outcomes on specific datasets, it is essential to note that such endeavours diverge from the focus of our investigation. Our goal is not to claim/create a SoTA result on a specific benchmark but to understand the potential for PEFT with carefully controlled experiments across a range of benchmarks. Our findings could of course be applied in conjunction with the tools employed by existing SoTA solutions, and would likely help to improve their performance going forward.
>
> **References:**
> 1. Loshchilov, Ilya and Frank Hutter. “Decoupled Weight Decay Regularization.” International Conference on Learning Representations (2017).

---

### Author Response · Authors · 2024-03-12
**Thank you for the review**

We would like to sincerely thank all the reviewers for their time and effort in providing constructive criticism of our work. We appreciate that reviewers recognize several merits of our paper such as _comprehensive experiments_, _well-structured and easy-to follow_, _good starting point for future benchmarks_ ,_sound motivation_, and _results serving as strong guidelines for the community_.

We have addressed the following main comments of the reviewers.

1. Including training details and comparison with existing SoTA (Reviewer mQqj)
2. Evaluating Domain-specific Pre-trained Models (Reviewer 7zWN)
3. Benchmarking only Classification and Generative Tasks (Reviewers 7zWN and ZQbg)

---

### Comment · Area_Chair_mk4u · 2024-03-13
**paper is open for discussions**

Dear Reviewers
The authors have submitted their rebuttal addressing the raised questions. The paper remains open for further discussion and engagement.

---

### Meta-Review · Area_Chair_mk4u · 2024-04-03

**Recommendation:** Accept (Poster)
**Confidence:** 5

**Metareview:**

All reviewers unanimously agreed to accept the work for presentation at the 2024 MIDL meeting. The authors are requested to revise the camera-ready version in accordance with the feedback provided in the rebuttal.

---

> ### Author Response · Authors · 2024-04-07
> **Regarding Oral Presentation Decision for Accepted Paper at MIDL 2024**
>
> Dear Area Chair mk4u
>
> Thank you very much for recommending an acceptance of our paper.
>
> Given the favourable evaluations, with scores of 4, 5, and 5, and particularly noting one reviewer's (reviewer 7zWN) recommendation for oral presentation, I am eager to know if the decision for oral presentations has been made.
>
> Looking forward to your reply.
>
> Thanks again and Warm Regards

---

### Decision · Program_Chairs · 2024-04-05

Accept (Oral)